# Quantitative Virus-Associated RNA Detection to Monitor Oncolytic Adenovirus Replication

**DOI:** 10.3390/ijms25126551

**Published:** 2024-06-14

**Authors:** Tereza Brachtlova, Jing Li, Ida H. van der Meulen-Muileman, Femke Sluiter, Willem von Meijenfeldt, Isabella Witte, Sanne Massaar, Ruben van den Oever, Jeroen de Vrij, Victor W. van Beusechem

**Affiliations:** 1Medical Oncology, Amsterdam UMC Location Vrije Universiteit Amsterdam, 1081 HV Amsterdam, The Netherlands; t.brachtlova@amsterdamumc.nl (T.B.); j.li2@amsterdamumc.nl (J.L.);; 2ORCA Therapeutics BV, 5223 DE ‘s-Hertogenbosch, The Netherlands; 3Cancer Biology and Immunology, Cancer Center Amsterdam, 1081 HV Amsterdam, The Netherlands; 4Cancer Immunology, Amsterdam Institute for Infection and Immunity, 1081 HV Amsterdam, The Netherlands; 5Department of Neurosurgery, Erasmus MC, 3015 GD Rotterdam, The Netherlands; 6ExoVectory BV, Biopartner 2 Building, 2333 CH Leiden, The Netherlands

**Keywords:** VA RNA I, virus replication monitoring, liquid biopsy, human xenograft tumor model, CAM tumor model

## Abstract

Oncolytic adenoviruses are in development as immunotherapeutic agents for solid tumors. Their efficacy is in part dependent on their ability to replicate in tumors. It is, however, difficult to obtain evidence for intratumoral oncolytic adenovirus replication if direct access to the tumor is not possible. Detection of systemic adenovirus DNA, which is sometimes used as a proxy, has limited value because it does not distinguish between the product of intratumoral replication and injected virus that did not replicate. Therefore, we investigated if detection of virus-associated RNA (VA RNA) by RT-qPCR on liquid biopsies could be used as an alternative. We found that VA RNA is expressed in adenovirus-infected cells in a replication-dependent manner and is secreted by these cells in association with extracellular vesicles. This allowed VA RNA detection in the peripheral blood of a preclinical in vivo model carrying adenovirus-injected human tumors and on liquid biopsies from a human clinical trial. Our results confirm that VA RNA detection in liquid biopsies can be used for minimally invasive assessment of oncolytic adenovirus replication in solid tumors in vivo.

## 1. Introduction

Oncolytic adenoviruses are a promising new class of anticancer agents. These viruses exhibit anticancer activity by selectively replicating in and killing cancer cells and by eliciting antitumor immune responses. Many different oncolytic adenoviruses are in development, most of which are derived from human adenovirus serotype 5 (Ad5). So far, they have already shown substantial preclinical and clinical efficacy [1,2]. Nevertheless, ongoing research aims to design more potent oncolytic adenovirus variants and validate their utility to treat a variety of cancers. One of the challenges in the development of oncolytic adenoviruses is the difficulty to obtain direct evidence of biological activity in vivo, in particular in clinical trials. Whilst tumor growth inhibition can sometimes be detected with imaging techniques, analysis of ongoing viral replication in the tumor requires tissue resection or taking biopsies. These invasive actions are often not possible at all or only deemed acceptable once. Therefore, a surrogate method is used where oncolytic adenovirus DNA copies are measured in the circulation via quantitative PCR (qPCR) [3,4,5]. This minimally invasive method can be repeated to monitor the presence of the virus in the body. Prolonged presence or an increase in copy number is considered indicative of oncolytic adenovirus replication. However, monitoring of adenoviral DNA copy number does not provide direct evidence of viral replication, because it does not distinguish between the detection of administered virus and produced progeny virus. Only the latter represents a consequence of replication. Furthermore, a qPCR in which a segment of the virus genome is amplified may also detect fragments of degraded virus DNA released from lysed cells. Release of these fragments from infected tumor tissue into the circulation could be delayed and thus represent virus-induced cell death that occurred before. Therefore, we sought for an alternative method that more accurately detects ongoing adenovirus replication. Here, we propose detection of Virus-associated RNA (VA RNA) as a marker for minimally invasive detection of oncolytic adenovirus replication.

VA RNAs are small non-coding RNAs transcribed from the adenoviral genome in infected host cells [6,7,8]. Human Ad5 expresses two VA RNA variants, VA RNA I and II. These short non-coding RNA species fold into structures very similar to precursor miRNAs. During replication, VA RNAs accumulate in high copy numbers. VA RNA I accumulates to the highest levels (about 10^8^ molecules per cell), while VA RNA II tops at around 10^7^ molecules per cell [6,7,8,9]. Expression of VA RNAs is important for efficient adenovirus replication. Earlier research showed that VA RNA I mainly enhances adenoviral replication by suppressing the interferon-induced protein kinase R (PKR) pathway. The PKR pathway can recognize dsRNA and induce phosphorylation of eIF2α, resulting in cessation of mRNA translation. The competitive inhibition of the PKR pathway by VA RNA I is thus important for efficient adenovirus gene expression [10,11]. VA RNAs are also processed in the cellular miRNA machinery to produce viral miRNAs (mivaRNAs) [7,12]. MivaRNAs derived from VA RNA I and II promote apoptosis and viral DNA replication, respectively, thereby supporting a productive virus life cycle [7,13,14,15].

We reasoned that because VA RNAs are produced in adenovirus-infected host cells and their abundance increases during replication [7,16,17,18], detection of VA RNA could provide a reliable assessment of ongoing adenovirus replication. In addition, since VA RNAs are processed in the miRNA biogenesis pathway [5,10] and miRNAs are sorted into exosomes and actively secreted from cells in these vesicles [19,20], we envisioned that perhaps also VA RNA-derived mivaRNAs could be released from virus-infected cells. If so, this could allow monitoring of ongoing oncolytic virus replication in a tumor by measuring secreted mivaRNAs using minimally invasive methods. Here, we investigated this, focusing on the most abundant VA RNA I. We show that VA RNA I is expressed in adenovirus-infected host cells; that VA RNA I abundance correlates with adenovirus replication and copy number; that VA RNA I abundance reflects the relative replication properties of different oncolytic adenoviruses; and that oncolytic adenovirus-infected cancer cells secrete VA RNA I-derived sequences associated with extracellular vesicles (EVs). By using a human non-small-cell lung carcinoma xenograft tumor model in fertilized chicken eggs, we demonstrate that after intratumoral administration of oncolytic adenovirus, VA RNA I sequences could be detected in the blood. Finally, we detected VA RNA I sequences in cerebrospinal fluid (CSF) samples of patients with glioblastoma multiforme (GBM) who were treated with an oncolytic adenovirus. Together, our findings suggest that measuring VA RNA copies in liquid biopsies could be used to assess replication of oncolytic adenovirus in tumor cells in vivo.

## 2. Results

### 2.1. VA RNA I Levels in Adenovirus-Infected Cells Are Replication and Dose Dependent

Previously, we analyzed VA RNA-derived small RNA species expressed in Ad5-infected cells by RNASeq. This revealed three main species; two expressed from the VA RNA I region and one from the VA RNA II region [6]. These most abundantly present VA RNA-derived species represent mivaRNAs with sequences from the 5′ terminal stem of VA RNA I (VA-RNAI-5p), the 3′ terminal stem of VA RNA I (VA-RNAI-3p) and the 3′ terminal stem of VA RNA II (VA-RNAII-3p), respectively [21]. To monitor VA RNA expression via RT-qPCR, we designed primers to detect these three species (Appendix A), using the stem–loop RT-PCR strategy developed for accurate quantification of small RNAs [22]. Appendix A illustrates the method for VA-RNAI-5p. In a pilot experiment, we infected HCT116 cells with oncolytic adenoviruses AdΔ24E3-U6 and AdΔ24E3-U6.pri-miR-1 [23]. At 8, 24 and 32 h post-infection (hpi), RNA was isolated and subjected to RT-qPCR for the three VA RNA-derived species. As can be seen in Appendix A, the highest copy numbers were detected using primers for VA-RNAI-5p. In addition, in contrast to the other VA RNA-derived species, VA-RNAI-5p could already be detected 24 h after infection. Therefore, the primer set detecting VA-RNAI-5p was chosen for use in all subsequent experiments. To investigate if the method detects expressed VA-RNAs or VA-RNA-encoding DNA sequences, we subjected RNA isolated from adenovirus-infected cells to the RT-qPCR assay with or without reverse transcriptase enzyme in the cDNA preparation step and/or after treating the isolated RNA with DNase I (Appendix A). VA-RNA sequences were not detected without reverse transcriptase; and signals were not reduced by DNase I treatment. We thus concluded that the VA-RNAI-5p RT-qPCR method detects primarily RNA species.

In a series of in vitro experiments, we measured VA-RNAI-5p levels in adenovirus-infected cancer cells. Because VA RNAs are expressed by replication-competent as well as replication-defective adenoviruses [24,25], A549 NSCLC cells were infected with either wild-type Ad5 or replication-defective adenovirus vector AdCMV-Luc [26] at a saturating multiplicity of infection (MOI). As is shown in Figure 1a, both viruses expressed VA-RNAI-5p, which was accumulating in infected cells. Ad5-infected cells showed a steep increase in VA-RNAI-5p copy numbers over the first 24 h after infection. This is consistent with the view that VA RNA expression increases with adenovirus genome replication. Next, we compared VA-RNAI-5p levels in A549 cells infected with Ad5 at different MOIs. This showed that VA-RNAI-5p levels accumulating in infected cells correlated with the MOI (Figure 1b; *R*^2^ = 0.9935, *p* < 0.01). Thus, VA-RNAI-5p quantitation reflects the extent of adenovirus infection under non-saturating conditions. To investigate if VA-RNAI-5p measurement can also be used to monitor ongoing viral replication, A549 cell cultures were infected with Ad5 at a low MOI and VA-RNAI-5p expression was measured daily for one week (Figure 1c). We observed that VA-RNAI-5p levels initially increased; then paused or even declined; before increasing further. The observed VA-RNAI-5p expression pattern in Ad5-infected cells suggested that two viral replication cycles occurred, where the pause represents the completion of the first lytic replication cycle. The time to complete the first cycle was MOI dependent (day 2 for MOI 10; day 4 for MOI 1; and day 5 for MOI 0.1). In control experiments, A549 cell cultures were infected with Ad5 or Ad-CMV-Luc and VA-RNAI-5p produced was measured for one week (Appendix A). While VA-RNAI-5p levels increased in Ad5-infected cultures, they remained constant in cultures infected with AdCMV-Luc. Thus, while expression of VA RNA I is not strictly replication dependent, detected levels do correlate with replication, confirming that VA-RNAI-5p quantitation could be used as marker for adenoviral replication.

To investigate whether VA-RNAI-5p expression correlates with adenoviral DNA copy number, A549 cells were infected with Ad5 at MOI 1 or MOI 100. Excess input virus was washed away and cells were cultured for 24, 48 or 72 h, to create samples with a range of adenovirus DNA and VA-RNAI-5p copy numbers. These samples were analyzed for adenovirus genome copy number by qPCR for Ad5 packaging domain sequences; and for VA-RNAI-5p expression by RT-qPCR. Of note, since cultures were depleted of input virus, only progeny genome copies were measured. Figure 1d shows that there was a strong correlation between adenovirus DNA and VA-RNAI-5p copy numbers (*R*^2^ = 0.92, *p* < 0.001). This suggests that VA-RNAI-5p RT-qPCR analysis could be used as alternative for adenovirus genome qPCR analysis, with the benefit of only detecting the result of replication.

### 2.2. Measuring VA RNA Expression Levels to Compare Adenovirus Replication Efficiencies in Cancer Cell Lines 

To investigate if VA-RNAI-5p copy numbers in infected cancer cells could be used as a quantitative readout method to compare adenovirus replication efficiencies, wild-type Ad5 and the oncolytic adenoviruses Ad5Δ24RGD [27] and ORCA-010 [28] were used to infect a panel of cell lines with differential permissiveness to adenoviral infection and replication [28,29]. After 32 h, when VA-RNAI-5p has accumulated to plateau values (see Figure 1), VA-RNAI-5p copy numbers were measured (Figure 2a). Ad5 produced VA-RNAI-5p copy numbers differing by approximately 2 orders of magnitude between A431 cells, known to poorly support adenovirus replication, and HCT116 and DU 145 cells, known to efficiently support adenovirus replication. Infectivity-enhanced Ad5Δ24RGD expressed more VA-RNAI-5p than Ad5 in A549, A431 and MDA-MB-231 cells, but not in the other three cell lines. Interestingly, ORCA-010, which is known to be the most potent of the three viruses in the comparison [28], reproducibly showed the highest VA-RNAI-5p copy numbers (*p* < 0.01 or <0.001 compared to Ad5 and Ad5Δ24RGD in all cell lines, except DU 145 cells). To confirm that the observed differences in VA-RNAI-5p copy numbers reflect differences in adenovirus replication efficiency, A549 cells were infected with Ad5Δ24RGD or ORCA-010 and adenovirus genome and VA-RNAI-5p copy numbers were quantified. As can be seen in Figure 2b, a strong correlation was observed. Together, these data show that RT-qPCR for VA-RNAI-5p can be used for comparative analysis of adenovirus replication in cancer cells, detecting effects of adenovirus genome modifications as well as intrinsic differences in host cell biology.

To test our hypothesis that VA RNA (upon processing into mivaRNA) is secreted by adenovirus-infected cells, we infected A549 cells with Ad5 and collected culture medium exposed to the virus-infected cells either during the first 24 h after infection or over the second 24 h after infection. Figure 2c shows that VA-RNAI-5p was detected in the culture supernatant. VA RNA concentrations were virus dose dependent and were higher in supernatant collected the second day after infection. The latter was expected, since replication-dependent VA RNA expression and processing for secretion will lag behind virus infection. At a higher MOI, a plateau in VA RNA concentration was reached, suggesting that the secretion machinery was saturated.

### 2.3. Detection of VA RNA in Extracellular Vesicles Produced by Adenovirus-Infected Cancer Cells

Almost all mammalian cells are known to shed EVs carrying a wide range of cargo, including RNA [30,31]. The RNA content of EVs is enriched for small RNAs, in particular miRNAs [32,33]. Since adenovirus VA RNAs are processed in the RNAi machinery to produce mivaRNAs [7,8,9], we reasoned that the VA-RNAI-5p molecules detected in the culture medium of adenovirus-infected cells were mivaRNAs loaded into EVs. To test this hypothesis, we infected HCT116 cells with two different oncolytic adenoviruses, i.e., AdΔ24E3-U6 and its miR-1-expressing derivative AdΔ24E3-U6.pri-miR-1 [23]. After allowing virus replication for 32 h, culture medium was harvested and EVs were isolated using two established methods, i.e., using a size exclusion column [33,34] or an iodixanol gradient [35]. Next, RT-qPCR for VA-RNAI-5p was performed on 100 ng RNA isolated from purified EVs. As can be seen in Table 1, cells infected with either virus shed EVs containing VA-RNAI-5p. AdΔ24E3-U6.pri-miR-1 produced lower levels of VA-RNAI-5p than did AdΔ24E3-U6. Because AdΔ24E3-U6.pri-miR-1 expresses very high levels of exogenous miR-1 in infected cells [23], we reasoned that the observation could perhaps be explained by competition between exogenous miR-1 and VA RNA for processing in the RNAi and EV loading machineries. Indeed, we could detect considerably more miR-1 in EVs from AdΔ24E3-U6.pri-miR-1-infected cells than from AdΔ24E3-U6-infected cells (Table 1). It is also shown in Table 1 that relatively large volumes culture medium were needed to isolate 100 ng EV-RNA and that these volumes were different using the two EV isolation methods. Thus, while the experiment served to confirm that secreted VA-RNAI-5p sequences are associated with EVs, it also showed that the applied EV purification methods need to be optimized for reproducible yield before they can be used to quantifyVA-RNAI-5p in liquid biopsies. 

### 2.4. Detection of Adenovirus DNA and VA RNA Copy Numbers in Oncolytic Virus-Injected Xenograft Tumors and Blood In Vivo

To investigate if measuring VA RNA copy numbers in the circulation could be used to monitor ongoing oncolytic adenovirus replication in a tumor in vivo, we employed a human NSCLC tumor xenograft model in fertilized chicken eggs [36]. H460-derived tumors were established on the chorioallantoic membrane (CAM) of fertilized chicken eggs and injected with oncolytic adenovirus ORCA-010. Starting one day after injection, on 6 consecutive days four embryos were sacrificed and tumor tissue and circulating blood were collected. Samples were processed for qPCR analysis for adenovirus genome sequences; for RT-qPCR analysis for VA-RNAI-5p copy numbers; and for infectious adenovirus titration. Of note, since different embryos were sacrificed for endpoint analysis at each day, we did not monitor ongoing virus replication longitudinally. Nevertheless, this analysis allowed recognizing patterns in viable virus titers and copy numbers over time. All analyses were performed on tumors and blood of control embryos that were not injected with virus were negative. Mean adenovirus DNA copy numbers in injected tumors remained rather constant throughout the duration of the experiment (Figure 3a). In contrast, VA-RNAI-5p copy numbers increased starting three days after ORCA-010 injection, suggesting active adenovirus replication (Figure 3b). Infectious oncolytic virus could be detected in tumors from the second day after ORCA-010 injection and live virus titers rose over the subsequent days (Figure 3c). From day 2 to day 5 after ORCA-010 injection, infectious virus titers and VA-RNAI-5p copy numbers increased, approximately 80-fold and 7-fold, respectively. VA-RNAI-5p levels and infectious units correlated (*R*^2^ = 0.821, *p* < 0.05). Thus, VA-RNAI-5p copy numbers were a better in vivo marker for ongoing adenovirus replication than adenovirus DNA copy numbers. The latter presumably include genome sequences of injected input virus that did not contribute to infection and replication and thus obscure the measurements, in particular in the first days after virus injection.

In the peripheral blood of NSCLC tumor-bearing embryos, adenovirus DNA was detected at all time points (Figure 4a). Copy numbers decreased a few days after injection and then increased again. VA-RNAI-5p copy numbers followed a different pattern (Figure 4b). They declined for 5 days, before reviving on the last day of measurement. Live infectious virus could be detected in the blood at only two time points, i.e., 1 day after injection and 4 days after injection (Figure 4c). At the other time points, virus titers were below the quantification threshold. This pattern could possibly reflect initial leakage of input virus from the injected tumor into the circulation followed by a first wave of progeny virus release three days later. The second peak infectious virus coincided with the observed increase in DNA copy number (Figure 4a). Thus, probably at least part of the virus genomes detected in the blood 4 days after ORCA-010 injection represents live progeny virus and thus confirms that ORCA-010 replicated in vivo. Unfortunately, this cannot be proven because input and progeny virus cannot be distinguished. Nevertheless, the detection of VA RNA every day in the blood suggested that ORCA-010 replication continued throughout the span of the experiment, even when live virus was undetectable in the circulation. It is, however, unknown when the detected VA-RNAI-5p copies were produced and how long it took before they reached the circulation. VA-RNAI-5p levels in blood did not correlate with VA-RNAI-5p levels in the tumor. They were several orders of magnitude lower, suggesting that only a minor proportion of VA-RNAI-5p was released from infected tumor cells into the circulation. Also, the patterns over time were different. Hence, while detection of VA RNA in the circulation seems useful to investigate if an oncolytic adenovirus is replicating in the tumor without the need to access the tumor, this cannot be used to quantify actual ongoing progeny virus production at the time of sampling.

### 2.5. Detection of Adenovirus VA RNA in Cerebrospinal Fluid of Patients with Glioblastoma Multiforme Treated with Oncolytic Adenovirus Ad5Δ24RGD

Finally, we performed a pilot experiment to investigate if VA-RNAI-5p could be detected in liquid biopsy material from humans undergoing experimental treatment of a solid tumor with an oncolytic adenovirus. For this, we employed samples from patients with GBM who were infused with Ad5Δ24RGD via convection-enhanced delivery to the tumor and the surrounding brain [37]. Previously, peripheral blood and CSF samples were already analyzed by qPCR for Ad5Δ24RGD DNA. In blood, the virus was not detected at any time point tested. In contrast, the oncolytic virus could be detected in CSF of many patients, often with a peak titer at 2 weeks after virus infusion [37]. Therefore, we analyzed CSF of selected patients from this trial for VA-RNAI-5p. We had CSF samples available from seven of the eight patients who received 10^10^ Ad5Δ24RGD particles. In one of these patients (#16), adenovirus DNA copy numbers remained low or undetectable, whereas the others had shown positive qPCR signals after adenovirus infusion [37]. Figure 5 shows the results of RT-qPCR analyses for VA-RNAI-5p on CSF sampled before virus infusion, during the first week after start of virus infusion (day 5 or 6 for different patients), after approximately 2 weeks and after approximately 1 month. As can be seen, VA-RNAI-5p could only be detected in a few cases. As expected, patient #16 was negative at all time points. This was also the case for CSF from patients 10, 11 and 18, in which adenovirus copies had been detected after 2 or 4 weeks [37]. Only the CSF of patient #12 in the first week and the CSF of patient #20 after 2 weeks tested clearly positive for VA-RNAI-5p. The results for patient #19 were inconclusive, because the preinfusion sample tested weakly positive. Thus, only for patients #12 and #20 evidence for Ad5Δ24RGD replication could be obtained.

## 3. Discussion

In this study, we investigated if VA RNA expression could be used as a means to obtain evidence for replication of oncolytic adenoviruses in a minimally invasive manner in vivo. To this end, we designed an RT-qPCR method that detects mature processed mivaRNA as well as its precursor VA RNA molecules; and selected VA-RNAI-5p as template for sensitive VA RNA detection. We found that VA-RNAI-5p copy numbers exhibited a steep increase during adenovirus replication in cancer cells. This replication-dependent expression makes VA RNA detection valuable as a method to assess active adenovirus replication. In contrast to measuring adenovirus DNA, it allows distinguishing between input virus and new progeny virus production.

At least in vitro, VA RNA measurement also allowed to quantitatively compare adenovirus replication efficiencies. VA-RNAI-5p levels correlated with multiplicities of infection used and when cell cultures were infected under non-saturating conditions adenovirus propagation could be monitored over multiple life cycles. Importantly, VA-RNAI-5p accumulated to higher levels in cancer cells that are highly susceptible to adenovirus infection and replication than in cancer cells that support adenovirus replication less well. In addition, VA-RNAI-5p copy numbers reflected known differences in oncolytic replication efficiencies between adenovirus variants. Infectivity-enhanced Ad5Δ24RGD produced more VA-RNAI-5p in several cell lines than unmodified Ad5; and ORCA-10, which is known to be more potent in killing cancer cells than Ad5Δ24RGD [28], produced the highest VA-RNAI-5p levels in all cancer cell lines tested. Together, clear correlations could be made between adenovirus replication and VA RNA expression.

The molecular difference between oncolytic adenoviruses Ad5Δ24RGD and ORCA-010 is that ORCA-010 carries a mutation in the E3/gp19k gene. This so-called T1 mutation was identified via bioselection for an adenovirus variant with potent antitumor activity from a randomly mutagenized Ad5 pool [38]. Characterization of Ad5/T1 showed that its mutant E3/19K protein permeabilizes the plasma membrane of host cells, thereby promoting early release of progeny virus. In contrast, total viral yield was not affected by the T1 mutation [38]. Hence, the existing view was that the T1 mutation affects only the final stage of the adenovirus life cycle, after completion of progeny virus production in host cells. Our observation that ORCA-010 produces more viral genome and VA RNA copies in host cells challenges this view. Notably, however, increased adenovirus DNA replication due to the T1 mutation not necessarily translates into an increased viral yield. Adenovirus genomes are produced in excess; DNA copy numbers accumulating in host cells are several orders of magnitude higher than infectious virus titers [39,40]. It is unknown if and to which extent the increased excess production of T1 mutant adenovirus genomes observed in our study contributes to the increased cytotoxicity of ORCA-010 in cancer cells [28].

Importantly, adenovirus-infected cancer cells were found to secrete VA-RNAI-5p into the culture medium. The amounts secreted appeared variable, making measurements on culture supernatant unreliable to quantify adenovirus replication in the cells that secreted VA-RNA-5p. We could confirm that secreted VA-RNA-5p was associated with extracellular vesicles. Purification of these vesicles reduced assay performance, because vesicle purification efficiency introduced an additional layer of variation. Nevertheless, presence of VA-RNAI-5p in supernatant allowed to assess that adenovirus replication was taking place in cells. This offered the opportunity to use liquid biopsies to detect ongoing adenovirus replication in cancer cells. To investigate if this was possible in vivo, we utilized the H460-CAM tumor model in which human NSCLC tumors are established on fertilized chicken eggs [36]. Upon oncolytic adenovirus ORCA-010 injection in H460-CAM tumors, rapidly increasing infectious virus titers were detected in dissected tumor tissues starting 2 days after injection. This coincided with increasing VA-RNAI-5p levels. In contrast, adenovirus DNA copy numbers were relatively constant over the six days analysis period. This showed that quantification of viral genomes was of limited relevance to assess virus progeny production. Measurement of VA RNA expression could be considered as a more appropriate alternative. Very low levels of VA-RNAI-5p could be detected in the blood of virus-injected tumor-bearing eggs. This confirmed that ongoing oncolytic adenovirus replication can be detected in liquid biopsies. Notably, the observed pattern suggested that circulating VA RNA is a qualitative rather than a quantitative marker of intratumoral adenovirus replication.

A pilot experiment on clinical trial specimens suggested that VA RNA detection in liquid biopsy samples could be used to confirm active oncolytic adenovirus replication in humans. However, the sensitivity of the current method appeared low. Only in samples from two out of six cancer patients who had shown presence of oncolytic adenovirus DNA [37] the presence of VA RNA could be detected. Although it cannot be ruled out that in the other four patients the administered virus did not substantially replicate; and the detected adenovirus DNA was thus mainly excess input virus or degraded virus DNA, we consider it more likely that the RT-qPCR method failed to detect less prominent adenovirus replication. Assay optimization is therefore warranted. One option for this could be to enrich for VA RNA by purifying extracellular vesicles. That will, however, require larger liquid biopsy volumes than were available for our study.

In conclusion, detection of VA RNA I shed by adenovirus-infected cells can be used to assess active adenovirus replication without direct access to the adenovirus-infected cells. This offers opportunity to obtain evidence of biological activity of oncolytic adenoviruses in vivo, via liquid biopsy sampling.

## 4. Materials and Methods

### 4.1. Cell Culture

A549 non-small-cell lung carcinoma cells, A431 epidermoid carcinoma cells, HCT116 colorectal carcinoma cells, MDA-MB-231 breast cancer cells and 911 adenovirus E1-complementing embryonic retinoblasts were grown in Dulbecco’s modified Eagle’s medium (DMEM) (Sigma-Aldrich Chemie NV, Zwijndrecht, The Netherlands). The SK-Mel-28 melanoma, H460 non-small-cell lung carcinoma and DU 145 prostate cancer cell lines were grown in RPMI-1640 medium (Sigma-Aldrich Chemie NV, Zwijndrecht, The Netherlands). All media were supplemented with 10% fetal calf serum (FBS; Gibco™, Fisher Scientific, Landsmeer, The Netherlands) and antibiotics (100 IU/mL penicillin and 100 µg/mL streptomycin) (Sigma-Aldrich Chemie NV, Zwijndrecht, The Netherlands). Cells were cultured at 37 °C in a 5% CO_2_ and 100% humidified atmosphere.

### 4.2. Adenoviruses

Viruses used in this study were wild-type human adenovirus serotype 5 (Ad5); replication-defective adenoviral vector AdCMV-Luc [26]; and oncolytic adenoviruses AdΔ24E3-U6 [23], AdΔ24E3-U6.pri-miR-1 [23], Ad5Δ24RGD [27] and ORCA-010 [28]. AdCMV-Luc was propagated on 911 cells; all other viruses on A549 cells. Virus infectious unit (IU) titers were determined by limiting dilution infection of 911 cells and hexon staining using the Adeno-X Rapid Titer Kit (Takara Bio Europe, Saint-Germain-en-Laye, France) 2 days after infection. In experiments, cell lines were infected at a MOI ranging from 0.1 to 100 IU/cell as indicated; and culture media were refreshed 3 h post-infection. Infected cells were cultured under standard conditions until analysis. Culture supernatant was collected from infected cells after 24 h; or culture medium was refreshed after 24 h and supernatant collected at 48 h after infection. Harvested supernatants were cleared by centrifugation before analysis.

### 4.3. Extracellular Vesicles Isolation and Analysis

Extracellular vesicles were isolated from the culture medium of adenovirus-infected cells at 32 hpi using two different isolation methods. The first method uses size-exclusion chromatography as previously described [33,34]. Briefly, the culture media were cleared of cells by four subsequent centrifugation steps, 2 times at 500× *g* for 10 min followed by 2 times at 2000× *g* for 15 min; before loading onto a Sepharose CL/2B size exclusion column. Sepharose CL/2B (GE Healthcare, Uppsala, Sweden) was washed in 0.32% citrate/PBS and then packed in a BD 10 mL syringe (San Jose, CA, USA) to a 10-mL volume. Six mL cleared supernatant was loaded onto the column, followed by collection of 0.5 mL gravity flow fractions. Fractions 9 and 10 containing extracellular vesicles [34] were collected and stored at −80 °C until RNA isolation.

The second method uses an iodixanol gradient, essentially as previously described [35]. Briefly, 150 mL collected culture supernatant was cleared by centrifugation at 150× *g* for 5 min, followed by centrifugation in new tubes at 3000× *g* for 20 min. Cleared supernatant was transferred to SW28 tubes (Beckmann Coulter, Brea, CA, USA), which were centrifuged at 100,000× *g* for 70 min to collect vesicles. Pellets were resuspended in 1 mL PBS and applied on top of a discontinuous iodixanol (Sigma Aldrich, Saint Louis, MO, USA) gradient (bottom to top: 2 mL 40%; 6 mL 25%; 2 mL 5%) in SW41 tubes (Beckmann Coulter). Layered samples were then centrifuged at 190,000× *g* for 4 h. After centrifugation, 11 fractions from the top of the gradient (1 mL each) were collected and stored at −80 °C until further use. Fraction 4 was used for experiments.

### 4.4. VA RNA and miR-1 Expression Analysis Using RT-qPCR

RNA isolation, cDNA preparation and RT-qPCR for human miR-1 expression analysis were performed as previously described [23]. For analysis of VA RNA expression, RNA was isolated from adenovirus infected cells, isolated extracellular vesicles or total blood using TRIzol (Invitrogen, Carlsbad, CA, USA) according to the manufacturer’s recommendations and from culture supernatant using the QIAamp DNA Blood Mini Kit (Qiagen, Hilden, Germany), according to the manufacturer’s protocol. Higher yields of small RNAs were achieved with addition of 0.4 mg/mL glycogen (Invitrogen) to the aqueous phase during isolation. RNA concentrations were measured using a NanoDrop Spectrophotometer (Thermo Fisher Scientific, Waltham, MA, USA). From 100 ng isolated RNA, cDNA was produced with the TaqMan MicroRNA Reverse Transcription kit (Applied Biosystems, Foster City, CA, USA) and a 10 μM VA RNA-specific stem–loop (SL) primer (primers for the different VA RNA species are described in Appendix A). Following the cDNA synthesis, the qPCR was run on 200 ng cDNA with a Roche LightCycler 480 qPCR system using 5× HOT FIREPol EvaGreen qPCR Mix Plus (no ROX) (Solis BioDyne, Tartu, Estonia), a VA RNA-specific forward primer and a reverse primer directed against a sequence in the SL primer (both 10 μM). Appendix A illustrates the method. The reactions were performed in technical duplicates and the VA RNA expression was presented as 2^−Ct^ values. Control experiments detecting VA-RNAI-5p without reverse transcriptase or with DNase I (Roche) treatment (Appendix A) showed that the RT-qPCR detects VA RNA, not the encoding DNA sequences.

### 4.5. Analysis of Adenovirus Genome Copy Number Using qPCR

DNA was extracted from infected cells or total blood using the QIAamp DNA Blood Mini Kit (Qiagen, Hilden, Germany), according to the manufacturer’s protocol. A qPCR specific for the adenovirus serotype 5 packaging domain was performed on a Roche LightCycler 480 system using 5× HOT FIREPol EvaGreen qPCR Mix Plus (no ROX) (Solis BioDyne, Tartu, Estonia) and custom primers 5′-GGAAGTGACAATTTTCGCGC-3′ and 5′-CCCGCGGCCCTAGACAAATAT-3′ designed to amplify a 161 nt sequence in the Ad5 packaging domain (nt200-nt360 of the human Ad5 genome; NCBI Reference Sequence AC_000008.1) that is commonly present in Ad5-derived adenovirus vectors and oncolytic viruses. The reactions were performed in technical duplicates and results are presented as 2^−Ct^ values.

### 4.6. In Vivo Experiments in a NSCLC-CAM Tumor Model

Establishment of human NSCLC xenograft tumors on the CAM of fertilized chicken eggs was performed essentially as described [36]. H460 NSCLC cells were grafted on the CAM on embryo development day (EDD) 6 and allowed to form tumors. Six days post-grafting (EDD12), ORCA-010 virus was injected intratumoral in 5 microliter PBS at a dose of 2.5 × 10^8^ IU per tumor. From EDD13 to EDD18, each day four embryos were sacrificed for collection of whole blood via venipuncture and for tumor excision. DNA was isolated from blood samples using the QIAamp Blood Mini Kit (Qiagen) essentially according to the manufacturer’s protocol, with extended incubation with proteinase K at 56 °C for 30 min. Tumor tissues were homogenized in 900 µL saline using a Precellys Evolution Tissue Homogenizer (Bertin Technologies, VWR International B.V., Amsterdam, The Netherlands, P000062-PEVO0-A) and 2 mL Hard Tissue homogenizing CK28 tubes (Bertin Technologies, VWR International B.V., Amsterdam, The Netherlands, P000911-LYSK0-A) following the manufacturer’s recommendations. Subsequently, samples were incubated with proteinase K and lysis buffer from the QIAamp Blood Mini kit for 1 h. As samples processed using the QIAamp Blood Mini kit contain DNA and RNA, a part of each sample could be used to isolate RNA using TRIzol as described above. Isolated DNA and RNA concentrations were measured using a NanoDrop Spectrophotometer (Thermo Fisher Scientific). qPCR detecting adenovirus copy numbers, cDNA preparation and qPCR analysis for VA-RNAI-5p levels were executed as described above. Titration of live virus from tumor and blood samples was performed using the Adeno-X Rapid Titer Kit in duplicate as described above.

### 4.7. Cerebrospinal Fluid from Patients with Glioblasoma Multiforme Who Were Injected with Oncolytic Adenovirus

We used samples from patients with recurrent GBM undergoing experimental treatment with oncolytic adenovirus Ad5Δ24RGD administered to the tumor and surrounding brain by convection-enhanced delivery (ClinicalTrials.gov Identifier: NCT01582516). CSF available from patient nrs. 10, 11, 12, 16, 18, 19 and 20 who received 10^10^ genome copies was collected, processed and stored as described [37]. Samples were collected before start of virus infusion, on day 5 or 6 after infusion, at approximately 2 weeks after start of infusion and approximately one month after start of infusion. RNA isolated from these samples was subjected to RT-qPCR analysis as decribed in Section 4.4.

### 4.8. Statistical Analyses

Statistical analyses were performed using GraphPad Prism (version 9.5.1; San Diego, CA, USA). Skewed data distributions were log transformed to achieve normality. Correlations were assessed using Pearson’s correlation. Data are presented as the median or the mean ± SD and statistical significance threshold is set at *p* < 0.05.

## 5. Conclusions

We found that VA RNA expression is indicative for ongoing adenovirus replication. In vitro, VA RNA expression correlates with adenoviral DNA copy number. Active shedding of VA RNA by adenovirus-infected cells offers the opportunity to measure VA RNA expression without direct access to the adenovirus-infected cells. We confirmed that this could be performed on peripheral blood in a preclinical xenograft lung tumor model and on CSF in a human brain cancer clinical trial. Thus, VA RNA detection in liquid biopsies allows minimally invasive evaluation of adenoviral replication in solid tumors in vivo. The complex kinetics of shedding from infected cells and biodistribution in bodily fluids confound quantitative analysis of adenovirus replication in vivo. Therefore, VA RNA detection in liquid biopsies should be considered primarily to obtain evidence of ongoing adenovirus replication, not for quantitative comparisons.

## Figures and Tables

**Figure 1 ijms-25-06551-f001:**
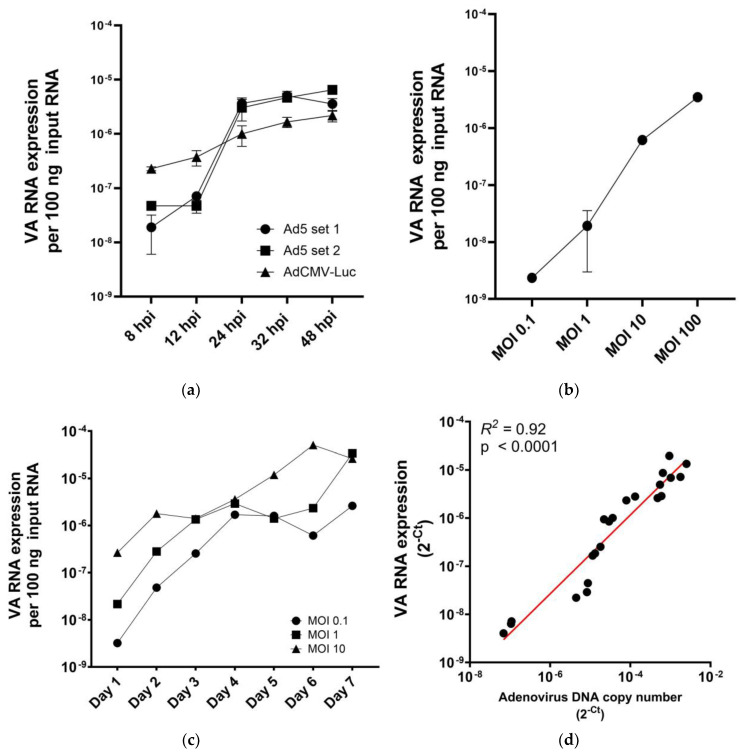
VA-RNA I expression in adenovirus-infected cells is time and MOI dependent. (**a**) Comparison of VA-RNA I expression in A549 cells transduced with replication-defective AdCMV-Luc (triangles; single experiment in triplicate) or infected with replication-competent Ad5 (circles and squares; two independent experiments in triplicate) at MOI 100 and analyzed over a course of 48 h. (**b**) Expression of VA-RNA I in A549 cells infected with Ad5 in a range of MOIs and analyzed at 32 hpi (*n* = 3). (**c**) VA-RNA I expression in A549 cell cultures infected with Ad5 at MOI 0.1 (circles), MOI 1 (squares) or MOI 10 (triangles) monitored over a 7-day period (single experiments). (**d**) Correlation between Ad5 DNA copy number and VA-RNAI-5p copy number in Ad5-infected A549 cells. Samples were prepared by infecting A549 cells with Ad5 at MOI 1 or 100; and harvesting cells for DNA/RNA isolation after 1, 2 or 3 days. Linear correlation was tested by calculating the Pearson correlation coefficient.

**Figure 2 ijms-25-06551-f002:**
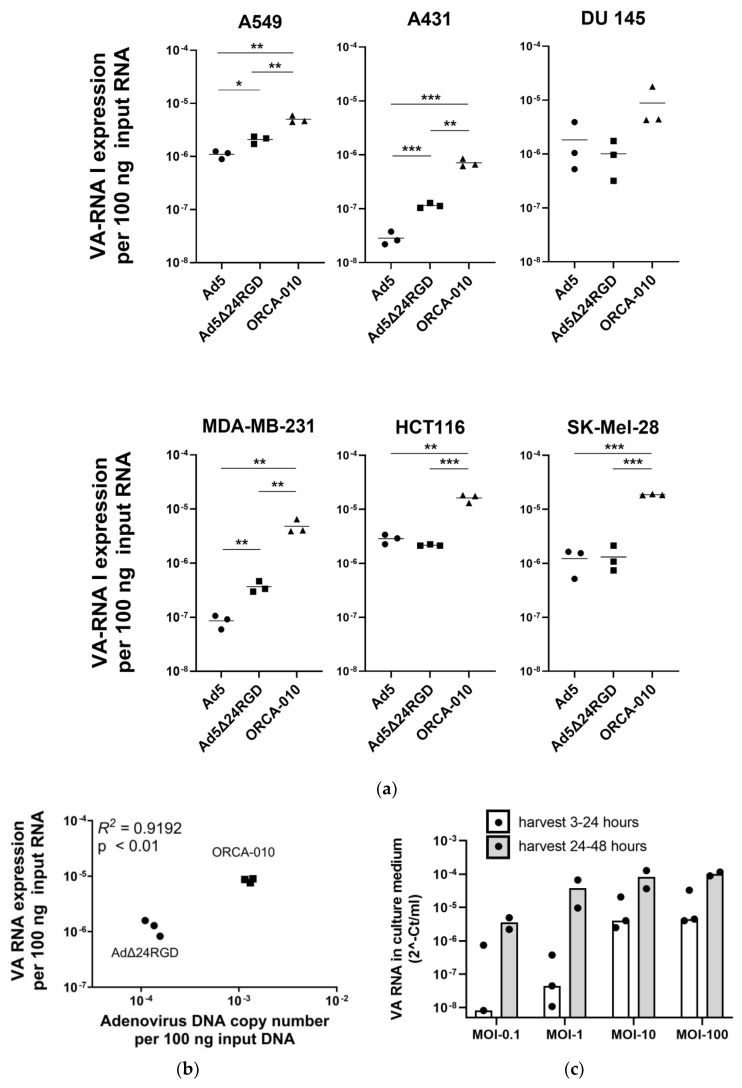
VA RNA levels in adenovirus-infected cancer cell lines and in culture supernatant. (**a**) VA-RNAI-5p level comparison in six human cancer cell lines infected with Ad5, Ad5Δ24RGD or ORCA-010 virus (*n* = 3, MOI 100, 32 hpi). Data shown are the results from individual infected cell cultures (dots; means of duplicate RT-qPCR analyses) with means per group (lines). Statistical significance of differences between viruses was tested by unpaired *t*-tests. *, *p* < 0.05; **, *p* < 0.01; ***, *p* < 0.001. (**b**) Correlation between adenovirus DNA copy number and VA-RNAI-5p expression in A549 cells infected with Ad5Δ24RGD or ORCA-010 (MOI 100). Data shown are the adenovirus DNA and VA-RNAI-5p copy numbers in independent virus-infected cell cultures. Linear correlation was tested by calculating the Pearson correlation coefficient. (**c**) VA-RNAI-5p content measured in culture supernatant of A549 cells infected with Ad5 at various MOIs. VA RNA accumulating in the culture medium on the first day (3–24 hpi) and on the second day (24–48 hpi) was measured separately. Data shown are individual results (dots) from three (3–24 hpi harvest) or two (24–48 hpi harvest) independent experiments performed in duplicate, with medians (bars).

**Figure 3 ijms-25-06551-f003:**
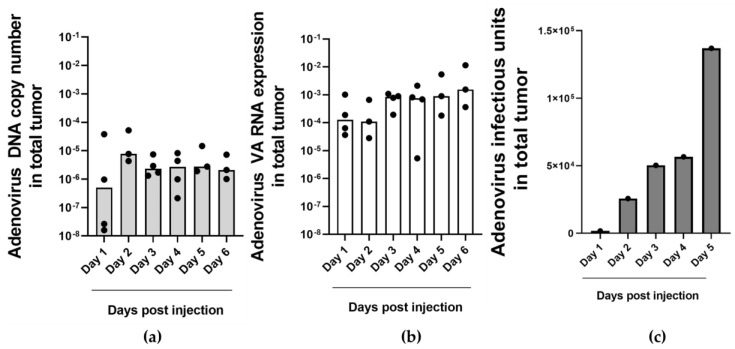
Adenovirus DNA and VA-RNAI-5p detection in adenovirus-treated NSCLC-CAM tumors. Adenovirus copy numbers (**a**), VA-RNAI-5p expression (**b**) and infectious adenovirus titers (**c**) in H460-CAM tumors that were injected with ORCA-010 and dissected on different days after virus injection. Data in (**a**,**b**) are the individual data (dots) of tumors analyzed in duplicate and medians (bars) per group; data in (**c**) are from single tumors analyzed in triplicate on each day.

**Figure 4 ijms-25-06551-f004:**
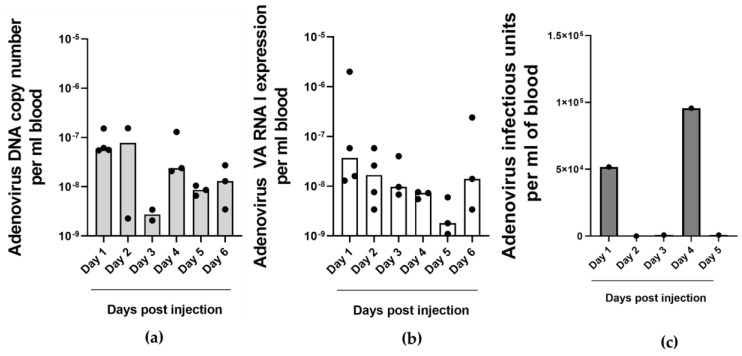
Adenovirus DNA and VA RNA I copy number detection in the blood of fertilized chicken eggs carrying adenovirus-treated NSCLC-CAM tumors. Adenovirus copy numbers (**a**), VA-RNAI-5p expression (**b**) and infectious adenovirus titers (**c**) in the blood of H460-CAM tumor-bearing eggs collected on different days after intratumoral ORCA-010 injection. Data in (**a**,**b**) are the individual data (dots) of tumors analyzed in duplicate and medians (bars) per group; data in (**c**) are from single tumors analyzed in triplicate on each day.

**Figure 5 ijms-25-06551-f005:**
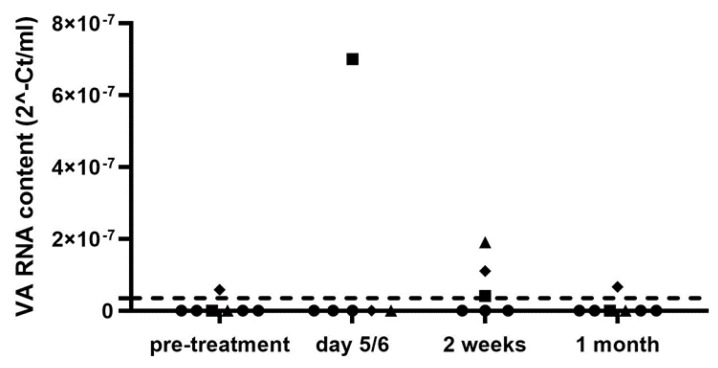
VA RNA detection in cerebrospinal fluid of patients with glioblastoma multiforme who were injected with oncolytic adenovirus Ad5Δ24RGD. CSF was collected before treatment and at multiple time points after infusion of 10^10^ viral particles into the tumor and surrounding brain via convection-enhanced delivery using implanted catheters [37]. The VA-RNAI-5p content was determined by RT-qPCR. Data shown are the individual data for all patient samples collected. The dashed line indicates the detection threshold in this experiment. Patients for whom at least one of the samples tested positive for VA-RNAI-5p are marked individually (squares, patient #12; diamonds, patient #19; triangles, patient #20). All other data are marked with dots. CSF samples were not available for patient #18 on day5/6 and for patient #10 at 2 weeks. Control qPCRs without RT step performed on the top 3 positive samples (CSF of patient #12 on day 5; and patients #19 and #20 after 2 weeks) were negative.

**Table 1 ijms-25-06551-t001:** VA RNA and miR-1 detected in RNA isolated from EVs produced by adenovirus-infected and control uninfected HCT116 cells. EVs were purified from culture medium using either size-exclusion chromatography or iodixanol gradient separation. Values are 2^−Ct^ per 100 ng isolated RNA. Culture medium volumes needed to isolate 100 ng RNA from the EV purification product are given.

	EV RNA Target	Control	AdΔ24E3-U6	AdΔ24E3- U6.pri-miR-1
**Size exclusion** **(0.86 mL medium)**	VA-RNAI-5p	Below detection level	2.47 × 10^−8^	1.29 × 10^−8^
	miR-1	3.06 × 10^−11^	2.40 × 10^−11^	9.41 × 10^−10^
**Iodixanol gradient** **(21.43 mL medium)**	VA-RNAI-5p	Below detection level	9.11 × 10^−8^	5.48 × 10^−9^
	miR-1	8.74 × 10^−11^	1.12 × 10^−10^	3.03 × 10^−6^

## Data Availability

The data presented in this study are available in this article and Appendix A.

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
