# Peer review of "Quantitative Virus-Associated RNA Detection to Monitor Oncolytic Adenovirus Replication"

_ijms, 2024, doi:10.3390/ijms25126551_

Round 1

Reviewer 1 Report

Comments and Suggestions for Authors

Title: Quantitative virus-associated RNA detection to monitor oncolytic adenovirus replication

·       Summary

In this manuscript, the authors reported a new approach for quantification of virus-associated RNA (VA-RNA-I) to monitor oncolytic adenovirus replication. The method can be applied to detect adenovirus in liquid samples. It is a valuable and critical method to increase to specifically target adenoviruses. In this manuscript, the authors well-organized data and well-written the manuscript. However, the authors should carefully consider reviewing next questions.

·       Major issues

1.      In figure 1a, the circles indicate two independent samples infected with replication-defective Ad5. Change one circle to square.

2.      The resolution of figure 2a is too low. Change the resolution.

3.      In figure 5, there is only data from cerebrospinal fluid sample. It may be required to add the number or size of tumor cells affected by adenovirus replication.

Reviewer 2 Report

Comments and Suggestions for Authors

In this study, Brachtlova et al. investigated the potentiality of VA RNA of being a non-invasive marker of oncolytic adenovirus replication. Oncolytic adenoviruses are currently under development to treat solid tumors and their efficacy rely on their active replication. So far, no proper non-invasive markers exist to unequivocally monitor viral replication. This study thus fits in this effort of identifying such marker. This work is an overall well-driven study that will be of interest to people working in this field. I have nonetheless some concerns the authors need to address before considering it for publication.

1) It is not clear to me in the Materials and Methods section whether DNAse treatment is sytematically performed in all the experiments shown in this work or whether it is only restricted to the Supplementary Figure 3. Here, DNAse treatment is critical to be able to discriminate between viral DNA and viral RNA. Can the authors clarify this point?

2) The authors demonstrated that both replication-competent and replication-defective viruses express VA RNAs intracellularly. I understood that the kinetic of expression of VA RNAs allows to discriminate between both viruses. The use of the replication-defective virus is restricted to the Figure 1. I think the authors should include it in all the in vitro experiments to be able to properly define VA RNAs as a marker of viral replication. 

3) The authors identified VA RNAs in the culture supernatant. Does the level of supernatant RNA correlate with the levels of intracellular RNA and viral replication?

Minor comment:

It would be more visual if the authors could include the legend of the Figure 1 a and c next to the panel. In its current form, we need to go to the legend to understand to what condition each line corresponds to. 

Round 2

Reviewer 1 Report

Comments and Suggestions for Authors

Dear. Authors,

Thank you for your work and they were enough to explain.

Best,

Reviewer 2 Report

Comments and Suggestions for Authors

The authors properly addressed my comments.